# The Effect of Ultrasound Treatment on the Structural and Functional Properties of *Tenebrio molitor* Myofibrillar Protein

**DOI:** 10.3390/foods13172817

**Published:** 2024-09-05

**Authors:** Xiu Wang, Xiangxiang Ni, Chaoyi Duan, Ruixi Li, Xiao’e Jiang, Mingfeng Xu, Rongrong Yu

**Affiliations:** 1School of Advanced Materials Engineering, Jiaxing Nanhu University, Jiaxing 314001, China; wangxiu@jxnhu.edu.cn; 2College of Life and Environmental Sciences, Hangzhou Normal University, Hangzhou 311121, China; 2023112010006@stu.hznu.edu.cn (X.N.); 2023210301138@stu.hznu.edu.cn (C.D.); 2023210301158@stu.hznu.edu.cn (R.L.); zjjxe1230@163.com (X.J.); 3The First Affiliated Hospital of Wenzhou Medical University, Wenzhou 325000, China

**Keywords:** *T. molitor*, myofibrillar protein, ultrasound treatment, structural properties, functional properties

## Abstract

The objective of this study was to explore the impacts of various ultrasonic powers (0, 300, 500, 700, and 900 W) on the structure and functional attributes of the myofibrillar protein (MP) of *Tenebrio molitor*. As the ultrasonic intensity escalated, the extraction efficiency and yield of the MP rose, while the particle size and turbidity decreased correspondingly. The reduction in sulfhydryl group content and the increase in carbonyl group content both suggested that ultrasonic treatment promoted the oxidation of the MP to a certain extent, which was conducive to the formation of a denser and more stable gel network structure. This was also affirmed by SEM images. Additionally, the findings of intrinsic fluorescence and FTIR indicated that high-intensity ultrasound significantly altered the secondary structure of the protein. The unfolding of the MP exposed more amino acid residues, the α-helix decreased, and the β-helix improved, thereby resulting in a looser and more flexible conformation. Along with the structural alteration, the surface hydrophobicity and emulsification properties were also significantly enhanced. Besides that, SDS–PAGE demonstrated that the MP of *T. molitor* was primarily composed of myosin heavy chain (MHC), actin, myosin light chain (MLC), paramyosin, and tropomyosin. The aforementioned results confirmed that ultrasonic treatment could, to a certain extent, enhance the structure and function of mealworm MP, thereby providing a theoretical reference for the utilization of edible insect proteins in the future, deep-processing proteins produced by *T. molitor*, and the development of new technologies.

## 1. Introduction

At present, with the continuous increase in the global population, the need for animal protein is also increasing day by day. However, the available animal protein resources are not sufficient to meet the daily needs of human beings. Alternative proteins, which are protein-rich foods used to replace animal resources, are currently important non-plant protein sources including insects. The consumption of insects is commonplace in many cultures globally and as an environmentally friendly source of protein, insect proteins have great potential for development. Compared with traditional animal protein, insects have less impact on environmental resources and emit fewer greenhouse gases during the rearing period. Moreover, it has been noted that the protein levels in multiple insects are as high as 35–61%, which is much higher than that of common plant proteins [1]. Modification is currently a major means of developing alternative proteins, which involves modifying proteins through various production methods to meet the global protein demand, which is estimated to nearly double by 2050 [2].

Ultrasound, as an emerging processing technology used in the modification process, has the advantages of safety, non-toxicity, and non-pollution. In recent years, it has been extensively utilized in the food machining sector, particularly for the improvement of protein structure and properties. Ultrasound primarily operates through the cavitation effect, generating bubbles. The rupture of these bubbles gives rise to transient high temperatures and pressures, high-energy shear micro-waves, and vortices in the cavitation zone [3]. High-intensity ultrasound propagating through proteins can cause compression and decompression of particles, causing changes in physicochemical properties and ultimately improving the quality of various systems [4]. At present, many investigations have demonstrated that the application of ultrasound can properly enhance the characteristics of proteins’ form and function. Chen et al. [5] found that an increase in the power of ultrasound significantly improved the stability and foaming activity, as well as the ability to contain water and oil, of natural barley proteins. Meanwhile, their secondary and tertiary structures had been altered. Studies on covalent reactions aided by ultrasound of chicken MP with polyphenols have demonstrated that the ultrasound-assisted ECG group had better antioxidant activity and digestive properties [6]. Amiri et al. [7] comparatively assessed the influence of ultrasound at high intensity on the function of MP in cows. They discovered that an increase in both time and power increased the sample’s ability to retain water and its gel strength, as well as enhanced its rheological properties.

*T. molitor* are generally referred to as their larvae, also known as mealworms. The main components of heat-dried *T. molitor* (NF) are proteins, fats, and chitin. And in 2018, EFSA was tasked by the European Commission with offering a scientific conclusion in response to a request to determine whether dried *T. molitor* is feasible as a novel food [8]. In actuality, one study found that the content of various amino acids in powdered *T. molitor* larvae was significantly high, including essential amino acids for the human body such as valine, isoleucine, and leucine [9]. It is precisely because *T. molitor* is rich in a variety of essential amino acids and nutrients that they are regarded as the third largest industrial insect after silkworms and bees and have a broad application prospect. MP is the most significant class of structural proteins in muscle and the main component of muscle protein, accounting for approximately 50–60% of the total protein in meat [10]. In addition, MP is essential for meat products’ emulsification, gelling, and other interfacial characteristics [11]. However, MPs are vulnerable to external factors when being processed and stored, which can lead to loss of flavor and quality or even deterioration of the meat. To date, most of the studies targeting *T. molitor* proteins (TMPs) have focused on the extraction process’s modification. To further improve the stability of *T. molitor* MP and to extend its use in the food field, there is a tremendous need to use different treatments to enhance its functional properties.

Thus, this study’s goal was to find out how ultrasound affected the physicochemical features and structure of *T. molitor* MP. The turbidity, average particle size, total sulfhydryl, and carbonyl content of the MP treated with various powers (0, 300, 500, 700, and 900 W) were compared. In addition, the specific changes in the MP produced by ultrasonic treatment were observed by sodium dodecyl sulfate–polyacrylamide gel electrophoresis (SDS–PAGE), spectroscopy, scanning electron microscopy (SEM), and so on. The present study aimed to offer novel concepts for *T. molitor* MP’s use in the food sector.

## 2. Materials and Methods

### 2.1. Materials

*T. molitor* (mature larval stage) was frozen at −20 °C for later use, ground into powder and sieved. Dithiothreitol (DTT) was acquired from Aladdin Reagent Co., Ltd. (Shanghai, China). The other analytical grade reagents used in this study were from Sinopharm Chemical Reagent Co., Ltd. (Shanghai, China).

### 2.2. Extraction of MP

*T. molitor* was washed, freeze-dried and stirred. A standard salt solution was configured, including 10 mmol/L Na_3_PO_4_, 0.1 mol/L NaCl, 2 mmol/L MgCl_2_, and 1 mmol/L EGTA, pH = 7.0. The extraction of the MP was completed by making minor adjustments to Chen et al. [6]’s method. The treated samples were mixed evenly with 4 times the volume of salt solution, then centrifuged (2000× *g*, 15 min, CF16RN, Tokyo, Japan) after being homogenized for 60 s to retain the precipitate. The above operation was repeated twice in total to obtain the crude MP of *T. molitor*. The crude MP was completely blended and homogenized for 60 s using four times the volume of 0.1 mol/L NaCl. The sample was thereafter centrifuged in a refrigerated centrifuge and filtered through 4 layers of gauze to retain the precipitate. This was repeated twice. HCl solution (0.1 mol/L) was added to restore the pH to 6.0. It was centrifuged again, and the precipitate obtained at this point was the final sample, which was placed at 4 °C for later use.

### 2.3. Ultrasonic Treatment of MP

The protein concentration of the sample was adjusted to 10 mg/mL, and 50 mL of suspension was placed in a 100 mL flat-bottomed conical flask. The MP sample was processed by a sonicator and the power was set at 300, 500, 700, or 900 W (SCIENTZ-II D, Ningbo, China). The probe was submerged in the sample solution, and then given a 30-min sonication under ice bath conditions, with 10 s intervals for every 10 s of sonication to prevent overheating [12]. After the ultrasound was completed, the MP suspension was stored at 4 °C for further analysis. A portion of the suspension was freeze-dried for 36 h, and the resulting sample powder was also refrigerated for later use.

### 2.4. Physicochemical Properties of MP after Ultrasonic Treatment

#### 2.4.1. Particle Size

The approach proposed by Amiri et al. [7] was used to determine the MP particle size. A static laser particle analyzer (DT-1202, DTI, Los Angeles, CA, USA) was used for particle size analysis based on the multimodal light-scattering method. Keep in mind that the particles were still scattered when the particle size was determined right after the sonication finished. The samples were diluted with phosphate buffered saline (PBS) buffer (10 mM, pH 7.0) to 1 mg/mL before particle size measurement.

#### 2.4.2. Total Sulfhydryl Content

First, Tris–Gly buffer was configured containing 86 mM Tris, 90 mM Gly, pH = 8. Next, 7.5 mg of the sample was dissolved in 9.5 mL of Tris–Gly buffer and 0.5 mL of 10 mM 5,5′-dithiobis (2-nitrobenzoic acid) (DTNB). Then, the above mixture was incubated at room temperature for 1 h and centrifuged for 10 min at 12,000× *g*. Lastly, at 412 nm, the absorbance value was determined and recorded as A_1_ (RF-6000, Shimadzu, Tokyo, Japan). Tris–Gly buffer was utilized as a blank control. The absorbance value was determined and noted as A_2_ [13]. The following formula was used to determine the MP’s total sulfhydryl content:Total sulfhydryl content (μmol/g) = (A_1_ − A_2_) × 14.706(1)
where 14.706 represented the molar absorbance coefficient of DTNB at 412 nm.

#### 2.4.3. Carbonyl Content

The carbonyl content was determined using 2,4-dinitrophenylhydrazine(DNPH) [14]. Pyrophosphate buffer was prepared containing 2 mM Na_2_P_4_O_7_, 10 mM Tris–maleate, 100 mM KCl, 2 mM MgCl_2_, and 2 mM ethylene glycol tetra acetic acid (EGTA), adjusting the pH to 7.4. The MP samples were dissolved in the buffer and homogenized for 30 s (T25, IKA Corporation, Stuttgart, Germany). Then, 1 mL of trichloroacetic acid (TCA, 20%) was taken and added to 200 μL of homogenate. After that, the samples were centrifuged at 12,000× *g* for 4 °C. The residue was washed using TCA (10%) and centrifuged again to remove the supernatant and retain the sediment. Next, 1 mL of DNPH solution was added and left to stand for 30 min at 37 °C in the absence of light. Then, 1 mL of TCA (20%) was added once more, and the residue was obtained by 5 min of centrifugation at 12,000× *g*. The precipitate was cleaned using 1 mL of a 1:1 ethanol/ethyl acetate (*v*/*v*) solution. The precipitate was dissolved in a mixture of PBS buffer (20 mM, pH 7.0) and guanidine hydrochloride (6 M), then incubated at 37 °C for 30 min. The supernatant was saved after centrifugation again. At 370 nm, the carbonyl content of the MP was measured.

#### 2.4.4. Turbidity

The determination of turbidity of the MP solution (1 mg/mL) was carried out by referring to Wei et al. [15]. The turbidity was measured at 660 nm using a spectrophotometer. Phosphate buffer (pH 7.0) was used as a blank control.

#### 2.4.5. Surface Hydrophobicity

8-Anilino-1-naphthalenesulfonic acid (ANS) was used to determine the surface hydrophobicity of the samples. In the beginning, the dilution of the freeze-dried samples was achieved by adding PBS buffer (pH 7.0) to adjust the protein solution concentration to 0.02, 0.04, 0.06, 0.08, and 0.1 mg/mL, respectively. Then, 3.5 mL of the sample was mixed with 60 μL of ANS solution (that was prepared to 8 mM using the above phosphate buffer). The excitation wavelength was set to 390 nm, and the emission wavelength was set to 470 nm. Plotting the relative fluorescence intensity against the protein concentration allowed us to determine the protein’s surface hydrophobicity, which was indicated by the curve’s initial slope [16]. 

#### 2.4.6. Emulsifying Properties

Emulsifying activity (EAI) and emulsifying stability (ESI) are important indicators for evaluating the emulsifying properties of proteins, which were determined with reference to the previous method [17]. An MP suspension (10 mg/mL) was taken and mixed with soybean oil 1:4 (*v*/*v*). The mixture was mixed well using a homogenizer set to 10,000× *g* for 30 s. Two homogenizations were performed. Then, 20 μL of the emulsion was quickly pipetted into 1 mg/mL SDS solution until the volume increased to 5 mL. The SDS solution by itself was used as the control. The absorbance value was determined at 500 nm and recorded as A_0_. This procedure was repeated after 10 min and the absorbance value was recorded as A_10_.
EAI (m^2^/g) = 115.5 × A_0_(2)
ESI (min) = A_0_/(A_0_ − A_10_) × 10(3)

### 2.5. Intrinsic Fluorescence Spectra

Firstly, the samples were diluted to 0.2 mg/mL with 10 mM PBS buffer (pH 7.0). The excitation wavelength was set to 295 nm, the slit width to 5 nm, and the wavelength range to 300–400 nm, and the emission spectra were recorded at 10 nm/s [18]. Each group of samples was scanned 3 times. 

### 2.6. Fourier Infrared Spectroscopy (FTIR)

To understand the secondary structural changes of proteins, FTIR analysis was performed on the MP samples [19]. The sample was taken and mixed with solid KBr powder (1 mg: 100 mg). A mortar and pestle were used to grind the samples to make them homogeneous, followed by pressing them into thin slices using a tablet press. The FTIR spectra of the MP samples were measured using a spectrometer (Perkin Elmer, Hopkinton, MA, USA) with the scanning wave number set between 4000–400 cm^−1^ for a total of 64 scans. Finally, Omnic IR software (version 9.0) was used to calculate the elemental content of the secondary structures.

### 2.7. Sodium Dodecyl Sulfate–Polyacrylamide Gel Electrophoresis (SDS–PAGE)

The MP solution was diluted to 2 mg/mL and mixed with a buffer solution (containing 62.5 mM Tris–HCl (pH 6.8), 2% SDS, 10% glycerol, and 0.01% bromophenol blue). A separation gel (10%) and stacking gel (5%) were prepared first. The above solution was heated at 90 °C for 5 min and then loaded into the gel for electrophoretic separation with 50 mM DTT to guarantee reduction conditions. A consistent voltage of 150 V was used to run the gels. After electrophoresis, the gel was stained for 30 min and then destained until the protein bands were clear. Coomassie brilliant blue G-250 (0.1%) was used for staining. The gel decolorizing solution contained 40% ethanol and 10% acetic acid [20,21]. 

### 2.8. Scanning Electron Microscopy (SEM)

Observations were made using an updated version of the method described by Shi et al. [22]. First, the MP samples were immersed in 0.2 M phosphate buffer (containing 2.5% glutaraldehyde, pH 7.2), and incubated for 3 h. The samples were then dehydrated with gradient ethanol (30%, 50%, 70%, 80%, 90% and 95%). After dehydration, the samples were washed three times with tert-butanol. Freeze-dried samples were vacuum-coated with a thin layer of gold–palladium prior to observation and fixed on the sample stage with conductive tape. Lastly, a scanning electron microscope (SU8010, HITACHI, Tokyo, Japan) with an accelerating voltage of 10 kV and a magnification of 5000× was used to examine the microstructure of MP.

### 2.9. Data Analysis

All sample groups were subjected to 3 independent experiments. Average values and standard deviations were calculated using IBM SPSS Statistics 27. Images were generated by the software Origin 2021. One-way analysis of variance (ANOVA) was used to analyze the statistical significance of differences between groups (*p* < 0.05). The lowercase letters in the figure indicate significant differences.

## 3. Results and Discussion

### 3.1. Extraction Efficiency and Yield

In this study, the percentage of MP mass to the total amount of protein was termed the extraction efficiency. Meanwhile, the mass of the MP sample obtained from 1 g of the *T. molitor* defatted powder was recorded as the extraction yield. The variation of efficiency and yield of MP under different ultrasonic powers is given in Figure 1. It could be observed that the increase in ultrasonic intensity resulted in a considerable improvement in both extraction efficiency and yield (*p* < 0.05). Among them, the extraction efficiency increased by 50%. The extraction yield increased from 0.202 g in the control group to 0.344 g, with a percentage increase of 70.8%. Even after increasing the ultrasonic power to 900 W, the yield and efficiency remained mostly unchanged (*p* < 0.05). These outcomes demonstrated that ultrasound did not offer additional benefits when excessive power was applied since the maximum yield was substantially reached. Ultrasound could promote tissue destruction through cavitation phenomena such as the formation of microjets [23]. This disruption could open the structure of the myogenic fibers and make the MP more exposed, thus effectively enhancing the MP extraction. However, the cavitation effect produced aberrant bubbles with excessively high ultrasonic power, which upset the bubble dynamics and deteriorated the cavitation effect instead [24]. Thus, a moderate ultrasound intensity could help improve the protein yield, while an excessive intensity did not have a more positive impact on the protein extraction process.

### 3.2. Particle Size

The shift in the MP’s mean particle size is depicted in Figure 2A. The average particle size of the protein decreased significantly from 469.3 nm to 165.67 nm as the ultrasonic strength was slowly raised to 700 W (*p* < 0.05). However, the subsequent intensification of power did not continue to diminish the particle size (*p* < 0.05). The lessening of the MP particle dimension might be related to cavitation forces, mechanical shear effects, and turbulence generated during the sonication process [25]. The particle size remained essentially unchanged despite the increase in ultrasonic power. This indicated that while ultrasound dispersed the protein, excessively high-intensity ultrasound instead caused aggregation of some of the isolated proteins. In a study on ultrasound-assisted or combined pH transfer methods, Silventoinen et al. [26] obtained similar conclusions, confirming that a reduction in particle size was observed after ultrasound at all pH values studied.

### 3.3. Total Sulfhydryl Group

Sulfhydryl group content is directly correlated with conformational changes and the degree of protein unfolding, which can represent alterations in the tertiary structure of proteins [27]. High-powered ultrasonography affects the functional properties of MP by changing the content of sulfhydryl groups. As seen in Figure 2B, the total sulfhydryl content of the blank control group was 53.63 μmol/g protein, and the ultrasonication group’s content was less than that of the control (*p* < 0.05). However, the difference was negligible until the ultrasonic power reached 500 W (*p* < 0.05). A decrease in protein particle size (Figure 2A) commonly also implies changes in the structure, which affects the content of sulfhydryl groups. Sulfhydryl groups were readily exposed to generate disulfide or sulfonic acid linkages during the processing of MP. This resulted in the loss of active sulfhydryl groups and produced protein disorder by destroying the intact structure and aggregative action of the proteins. The above reasons caused the total sulfhydryl group content to decrease ultimately [28]. A similar result has been observed by Li et al. [29]. On the one hand, this was probably due to the formation of disulfide bonds. On the other hand, it might be because the high-activity free radicals formed hydrogen peroxide, which oxidized surface-exposed sulfhydryl groups into intramolecular or intermolecular disulfide bonds [30]. This result was consistent with that of Zhang et al. [31], who explored the effects of ultrasound on low-salt *Hypophthalmichthys molitrix* surimi. 

### 3.4. Carbonyl Group 

The carbonyl content of MP increased from 1.11 nmol/mg to 3.03 nmol/mg, an increase of approximately 1.73 times, during the process of the ultrasound intensity increasing to 900 W, as illustrated in Figure 2C (*p* < 0.05). In general, the carbonyl content of proteins is related to the degree of oxidation, and the degree of protein oxidation increases with an increasing carbonyl concentration [29]. It has been demonstrated that amino acid residues such as cysteine are susceptible to oxidation by reactive oxygen groups to produce carbonyl derivatives [14]. This indicated that ultrasonic treatment promoted the occurrence of MP oxidation and exposed the reduced amino acid side chains to free radicals and oxidized them to carbonyl groups. This was mainly because the cavitation effect produced by ultrasound could bring about a local high temperature, which caused variations in the protein’s structure and created denaturation [32]. Zhang et al. [33] also reported a similar phenomenon under high-intensity ultrasound treatment, where the carbonyl content of pork MP significantly increased. In summary, these outcomes showed that ultrasound with high intensity greatly increased the susceptibility of MP to free radical attack [17], which promoted the production of carbonyl groups and MP oxidation. A decrease in sulfhydryl content is another important indicator of protein oxidation [34]. The decrease in sulfhydryl content mentioned above further confirmed this point. This oxidation could promote the stability and density of the MP gel structure to a certain extent.

### 3.5. Turbidity

The impact of ultrasound on MP turbidity is depicted in Figure 2D. Turbidity reflects the degree of aggregation of protein molecules [35]. The lower the turbidity value, the lower the degree of protein aggregation. Observing the pictures reveals that the turbidity of MP always maintains a decreasing trend, and with the enhancement of ultrasound, the degree of decline reached 48.5% (*p* < 0.05). This suggested that the protein sample’s particle size shrunk; the ultrasound dispersed the larger aggregates to form smaller particles and therefore had less light scattering [25]. This further supported the degradation of the protein structure by ultrasonic treatment and correlated with the earlier reduction in particle size (Figure 2A). In addition, the results suggested that the high shear force generated by ultrasonic waves might also disrupt the interactions between protein molecules, thus reducing the degree of aggregation. It is noteworthy that similar conclusions have been obtained by others. Chen et al. [36] suggested that the shear effect’s breakdown of the MP’s initial structure was the cause of the drop in turbidity seen in response to an increase in ultrasonic frequency. Huang et al. [27] obtained the same phenomenon in their experiments with yellow powdery mildew protein, and discovered a negative correlation between the intensity and the turbidity.

### 3.6. Surface Hydrophobicity

Surface hydrophobicity is an important physicochemical property of protein molecules, which is mainly manifested in the repulsion of water molecules. It represents the arrangement of hydrophobic residues on the molecule’s surface, and has a certain impact on the conformation and function of proteins [16]. As shown in Figure 3A, ultrasonication obviously improved the surface hydrophobicity (*p* < 0.05). During the process of ultrasound rising to 500 W, the surface hydrophobicity increased by about 2.35 times compared to the control group (*p* < 0.05). Surface hydrophobicity reduced the interaction forces between particles, reduced the agglomeration, and made it easier to form stable aggregates. This would ultimately lead to the adsorption and aggregation of water molecules, thereby reducing turbidity, which was consistent with the previous results. This was principally because the high-intensity ultrasound’s shock waves, turbulence, and shear spread the protein molecules, exposing the hydrophobic amino acids that had previously been concealed inside them [37]. Interestingly, when 700 W of ultrasonic power was applied, the hydrophobicity of MP instead decreased slightly, lower than the surface hydrophobicity at 500 W (*p* < 0.05). This indicated that when the ultrasonic power was too high, it might cause some protein molecules to experience denaturation and repolymerization with the help of hydrophobic interactions. In this way, some hydrophobic residues were buried again, which eventually caused the proteins’ surface hydrophobicity to decrease [38].

### 3.7. Emulsifying Properties

The emulsifying property describes the capacity of molecules at the interface between oil and water to be taken into the oil–water system to form a homogeneous emulsion and to keep the emulsion system in a stable state [39], which is mainly determined by two parameters, EAI and ESI. The former corresponds to the protein’s capacity to be adsorbed during emulsification and represents the maximum adsorption of oil by the entire myofibrillar protein system. The latter denotes the ability of proteins to remain emulsified by staying in contact between the oil and water in the absence of water and oil separation [17]. As could be seen in Figure 3B, ultrasonication had a favorable impact on ESI and EAI, both of which were significantly improved (*p* < 0.05). The EAI increased from 93.88 m^2^/g to 212.69 m^2^/g, an increase of approximately 2.26 times. And the ESI increased by about 6 times. Previous studies showed that the particle size (Figure 2A) and turbidity (Figure 2D) of the proteins were significantly decreased (*p* < 0.05). This was precisely because ultrasound altered the MP’s composition, lowered the particle’s size and raised its surface-to-volume ratio. This helped in the formation of the interface layer, and made it easier to cover the periphery of the oil droplet [40]. Ultrasonication also promoted the solubilizing effect, which contributed to preserving the oil–water interface’s stability. In addition, as the sonication intensity increased, more hydrophobic residues were exposed, which enhanced the surface hydrophobicity of the protein. Moreover, it improved the transfer rate and adsorption ratio at the oil–water interface, as well as the hydrophilic–lipophilic balance, which ultimately led to a further increase in the emulsifying properties [41]. However, it can also be visualized that at first both EAI and ESI increased dramatically with the change in ultrasound power. Upon attaining 700 W of ultrasonic power, the enhancement of emulsification performance was not significant and slowed down obviously. This indicates that excessive ultrasonication might cause the aggregation of some proteins, destroying their adsorption capacity and the maintenance of the stable state.

### 3.8. Intrinsic Fluorescence Spectroscopy

Intrinsic fluorescence spectroscopy can be employed to keep an eye on proteins’ tertiary structures. *T. molitor* MP contains a variety of aromatic amino acids (Trp, Tyr, and Phe). When exposed to 280 nm excitation light, critical fluorescence can be generated based on protein folding, especially for Trp [42]. When the conformation of a protein changes due to external factors, the interaction of the aromatic amino acids with the surrounding groups and the microenvironment in which they reside will also change accordingly. This can cause changes in fluorescence intensity, thus playing an important role in monitoring the tertiary structure of proteins. As seen in Figure 4A, the control without sonication had the maximal absorption value near 330 nm, and with the steady increase in power, the fluorescence intensity dropped significantly. This indicated that sonication changed the tertiary structure or aggregation state of the MP. In addition, a slight redshift in λmax of the MP could be found. According to a previous study, it is known that when Trp is shifted to a larger wavelength (redshift), this means that Trp is immersed in a hydrophilic environment [18]. One possible reason is that high-intensity ultrasound induces protein unfolding. This results in more chromophores such as Trp being visible on the protein molecule’s surface in a hydrophilic environment [43], making the environment around the amino acid more polar. When the chromophore is exposed to solvent, a solvent quenching effect occurs, thereby reducing the fluorescence intensity [44]. And the higher the ultrasound power, the higher the degree of exposure. Xiong et al. [45] obtained similar results. They concluded that it might be the result of the difference in energy transfer efficiency between Trp and Tyr, as well as the exposure of the chromophore groups to solvents. 

### 3.9. FTIR

FTIR is usually used to study the secondary structure of proteins, including α-helix, β-sheet, β-turn, random coil, and changes in functional groups [46]. Different structural forms will correspond to various absorption peaks, as shown in Figure 4B. According to the infrared spectrogram, the summits’ positions and shapes did not change much, indicating that the ultrasonic treatment did not lead to the thorough destruction or formation of protein functional groups [47]. In addition, it can be observed from the figure that the intensity of the absorption peaks at 2962 cm^−1^ and 1124 cm^−1^ was slightly weakened with the increase of the ultrasonic power. This mainly corresponded to the stretching and bending vibration of methyl (-CH_3_) and methylene (-CH_2_) in the MP, indicating that the content of methyl and methylene was reduced by ultrasound assistance. Observing the FTIR spectrum, it was also found that the control shows a distinct vibrational band at 1526 cm^−1^, which was confirmed to be an amide group (C-N band) [48]. The presence of a carbonyl group is shown by the distinctive signal at 1681 cm^−1^ (C=O band). The amide I band (1600–1700 cm^−1^) and the amide II band (1500–1600 cm^−1^) are associated with these two peaks, respectively. They are the characteristic absorption peaks of proteins, among which the amide I band can be applied to the analysis of proteins’ secondary structures. In addition, a broad and strong absorption peak can be seen in the range of 3200–3600 cm^−1^ regardless of whether it had been treated with ultrasound or not, which was mostly connected to the -OH’s stretching vibration and the hydrogen bonding in the protein. This peak was slightly blue-shifted and then red-shifted with the gradual increase of the sonication power. The changes in the absorption peaks’ positions near 724 cm^−1^ and 1186 cm^−1^ similarly revealed the variations in the C-H group’s vibration.

From the FTIR spectrum, as the ultrasound power climbed, the MP’s amide I band shifted, indicating that the secondary structure of the proteins changed as well. Specifically, the α-helix, β-sheet, β-turn, and random coil of proteins were represented by several peaks between 1600 and 1700 cm^−1^. Figure 4C shows the content of various secondary structures in the MP at different ultrasonic powers. During the gradual increase of ultrasound power up to 700 W, the content of the protein α-helix and β-sheet both decreased continuously, while the content of the random coil increased (*p* < 0.05). This proved that the cavitation effect and mechanical stress generated by the ultrasound disrupted the rigid structure of the protein molecules, making the structure more flexible and looser. This increased the flexibility of the protein molecules [49]. However, it could also be observed that when the ultrasound power was too high, it instead caused a large increase in the β-sheet content. The α-helix and β-sheet are mainly stabilized by hydrogen bonds within and between peptide chains, respectively. On the one hand, since overly intense ultrasound further caused damage to the hydrogen bonds within the peptide chains, this promoted the reconstruction of the hydrogen bonds between the peptide chains, which converted part of the α-helix into the β-sheet structure [50]. On the other hand, the relative content of β-sheet is also related to the hydrophobicity and stability of proteins [51]. When the ultrasonic power was too high, the proteins were denatured, bound to each other, and formed small aggregates. The MP was less exposed to the hydrophobic regions inside the molecule, promoting the rise of β-sheet content. This was in line with the changes in the average particle size of the MP mentioned earlier. The variation in random coil content was opposite to that of the β-sheet, showing an upward trend first and then decreasing. All of these changes indicated that sonication disrupted the tight structure of the protein, making it more disordered and looser. 

### 3.10. SDS–PAGE

Figure 5 illustrates how SDS–PAGE is used to examine the impact of sonication on the arrangement of molecular weights and energy band intensity of MP. Protein bands of 27, 35, 45, 85, and 180 kDa were shown at all powers, corresponding to MHC, paramyosin, actin, tropomyosin, and MLC, respectively [29]. In comparison to the control, the distribution of protein bands did not change significantly. This was comparable to the results of Zhang [43] and Wang et al. [52], who showed that the electrophoretic pattern of the protein as well as the MP’s profile barely changed as the ultrasound’s intensity varied. Although the subunit composition of the MP was essentially the same, the bands of paramyosin and MHC became narrower and fainter as the power rose. These changes implied that sonication reduced the molecular weight of both proteins, causing them to undergo a slight degree of degradation. This phenomenon might be caused by turbulence, microfluidics, shear, and cavitation [53].

### 3.11. Scanning Electron Microscopy (SEM)

The MP gel’s microstructure was examined, as seen in Figure 6. The MP gel without ultrasonication showed large pores, probably due to irregular aggregation between proteins and a low degree of network cross-linking [54]. After treatment with ultrasound, the MP gels showed a more aggregated and dense structure with gradually decreasing pores. The decrease in protein size (Figure 2A) and sulfhydryl content (Figure 2B) could be the cause of this phenomenon. At the same time, due to the exposure of more hydrophobic groups by ultrasound, more cross-linking between the molecules was formed through hydrophobic interactions and disulfide bonding, resulting in a more homogeneous gel network [55]. However, the increase in the flexibility of the proteins might also promote the homogeneity and density of the gel network [56]. In addition, some dispersed and tiny particles were observed at ultrasound powers of 500 and 700 W, respectively. It demonstrated that the turbulence, microfluidic forces, and cavitation produced by ultrasound caused the MP to fracture to varying degrees, producing partially sparse structures. Following another increase in ultrasound intensity, these small particles nearly vanished, aggregating with each other. These results are consistent with the variation in castor (*Crotalaria juncea* L.) protein isolates [57]. 

## 4. Conclusions

This study comprehensively investigated the specific effects and mechanisms produced by different ultrasound intensities on *T. molitor* MP. The results showed that ultrasonic treatment effectively increased the extraction yield and efficiency of MP compared to the un-sonicated control group. Ultrasonication could also reduce protein turbidity and the average particle size and improve the surface hydrophobicity of MP through cavitation effects, microfluidics, turbulence, and so on. The EAI and ESI were significantly improved. In addition, the samples treated with ultrasound showed a significant decrease in total sulfhydryl content and an increase in carbonyl content, promoting MP oxidation. With the help of FTIR and intrinsic fluorescence spectroscopy, the secondary/tertiary structure of the sonicated MP was analyzed. This once again confirmed that ultrasound caused the protein structure to undergo unfolding and significant changes, becoming looser and more flexible. SEM images proved that a denser and more consistent gel network structure was formed. The results of SDS–PAGE electrophoresis demonstrated that ultrasonication did not change the basic subunit composition or the distribution of MP. In summary, these results provide a theoretical basis for the application of ultrasound in the production of edible insect MP. Further research is needed on the combination of ultrasound and other technologies to enhance the applications of insect proteins.

## Figures and Tables

**Figure 1 foods-13-02817-f001:**
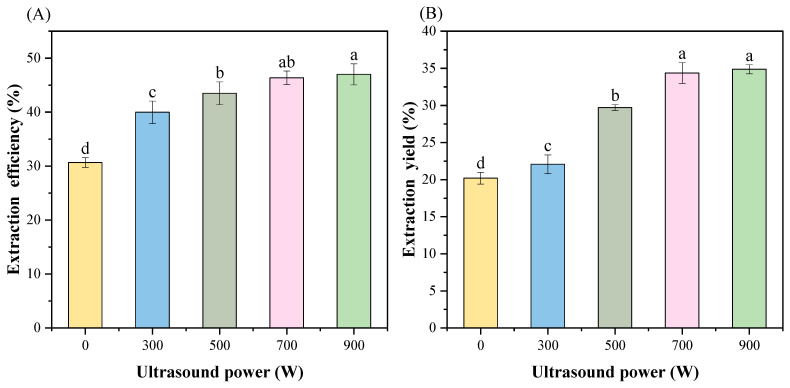
Effect of different ultrasound powers on MP extraction efficiency (**A**) and yield (**B**) of *T. molitor*. Different letters represent significant differences (*p* < 0.05).

**Figure 2 foods-13-02817-f002:**
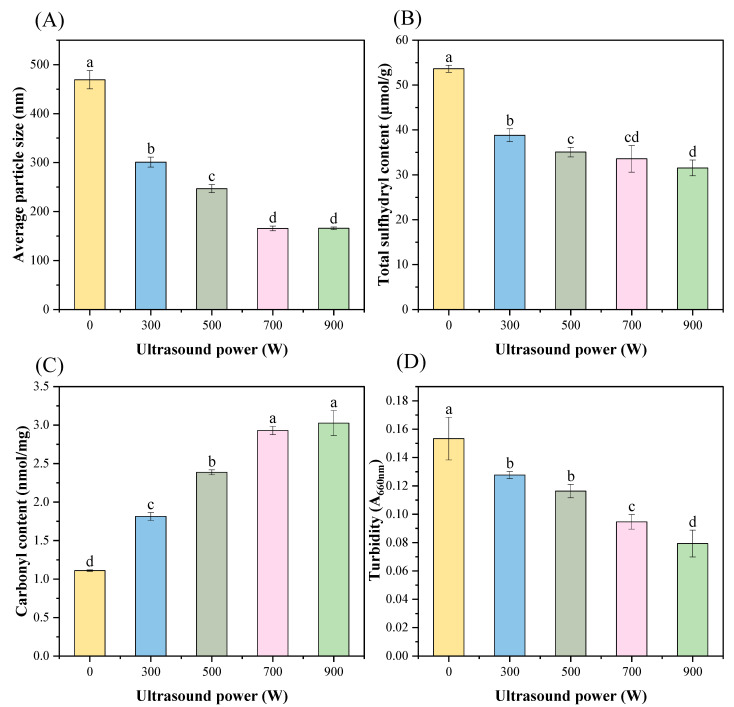
Effect of different ultrasound powers on the particle size (**A**), total sulfhydryl content (**B**), carbonyl content (**C**), and turbidity (**D**) of *T. molitor* MP. Different letters represent significant differences (*p* < 0.05).

**Figure 3 foods-13-02817-f003:**
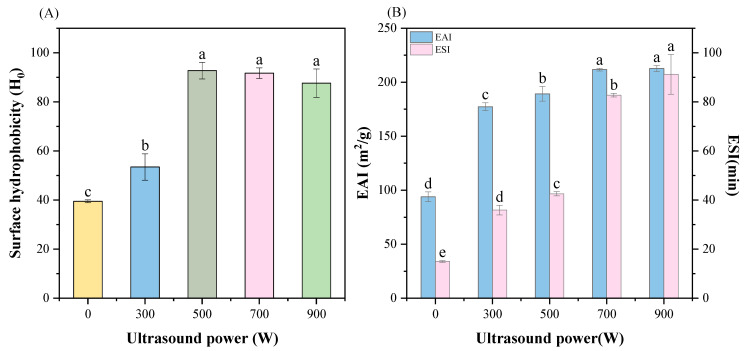
Effect of different ultrasound powers on the surface hydrophobicity (**A**) and emulsifying performance (**B**) of *T. molitor* MP. Different letters represent significant differences (*p* < 0.05).

**Figure 4 foods-13-02817-f004:**
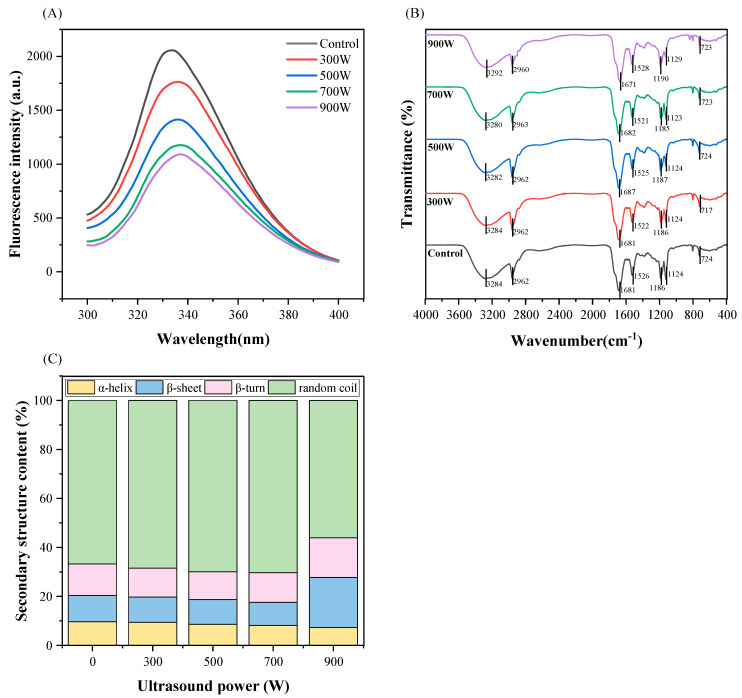
The effect of different ultrasound powers on the intrinsic fluorescence spectrum (**A**), FTIR spectrum (**B**), and secondary structure content (**C**) of *T. molitor* MP.

**Figure 5 foods-13-02817-f005:**
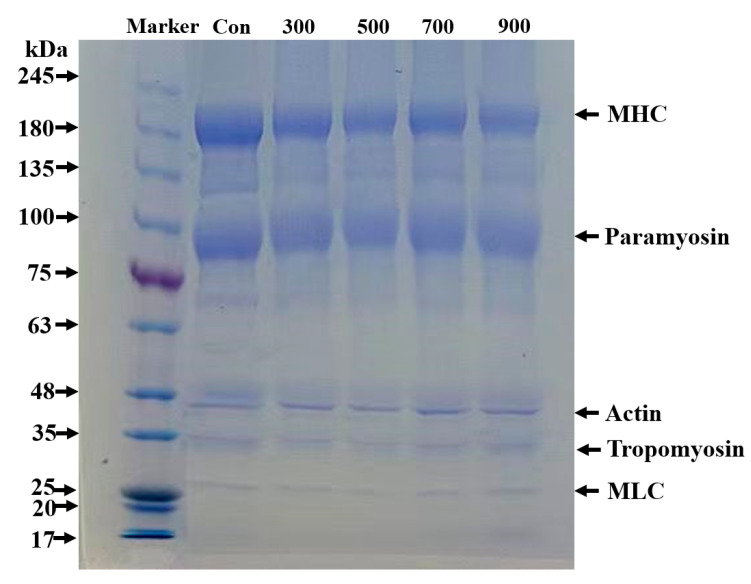
Effect of different ultrasound powers on the protein composition of *T. molitor* MP.

**Figure 6 foods-13-02817-f006:**
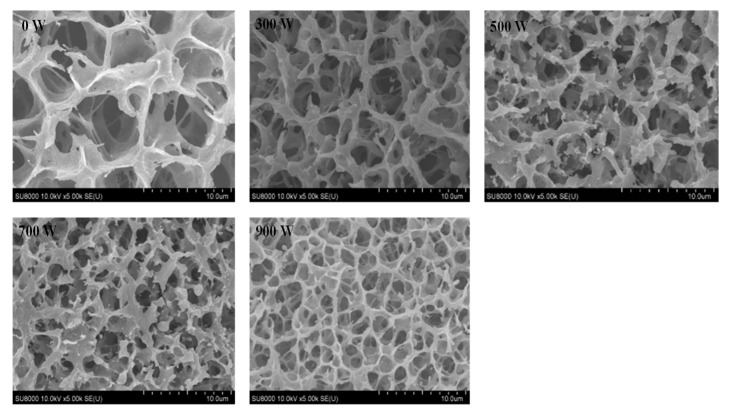
Effect of ultrasound powers on the microstructure of *T. molitor* MP.

## Data Availability

The original contributions presented in the study are included in the article, further inquiries can be directed to the corresponding author.

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
