# Peer review of "The Effect of Ultrasound Treatment on the Structural and Functional Properties of Tenebrio molitor Myofibrillar Protein"

_foods, 2024, doi:10.3390/foods13172817_

Round 1

Reviewer 1 Report

Comments and Suggestions for Authors

The manuscript “The effect of ultrasound treatment on the structural and functional properties of Tenebrio Molitor myofibrillar protein” exploited the alternative protein source. Although the effects of ultrasound on the proteins are well known, current paper also followed the same trend. No new information is obtained from the current manuscript. The methodology needs to improve as it lacks certain details. Also, discussion is not up to the mark in all section authors have explained the same reason, which is cavitation effect. This manuscript needs extensive editing as following comments:
###All the changes should be highlighted with different color for each author and line number should mentioned while commenting####

1.      As authors tried to explore this as an alternative protein, but no nutritional profile has been given. It must that author should determine amino acid composition of the selected sample. Otherwise, this paper reflects no new information.

2.      I cannot point out every mistake, but those have not used the appropriate space between the wors. Examples can be seen in the captions of Figures.  Similarly, 1 or 2 or whatever no space between the number and unit. Line 227-228 and Figure 2.

3.      Abstract is very poor, need to present the major finding of the results such as yield, EAI/ESI values, etc. should be given. It must stand alone. It does not reflect that at what treatment, what functional properties were improved.

4.      I suggest to remove term non-meat protein source at line 29. This is also animal protein.

5.      How T. molitor can be of analytical grade. Molitor should be molitor based on the scientific name. Check this throughout the manuscript.

6.      All methods should detailed with the concentrations of the sample.

7.      Section 2.3 should be explained in detail, as authors would like to present new information. No information about the concentration of the sample is given. It said only 100 ml conical flask, but how much solution was used.

8.      It is not clear when the author used liquid or solid sample. As mentioned in 2.4.1 and then again used powder in 2.4.7. When did the author freeze-dried the samples?? Explain.

9.      14.706 should explained.

10.  In line 111, authors used DNTB but no full form and then again in line 119 and 120, authors have mentioned the full form for DNPH but not for EGTA, all are known chemicals. OS authors should be consistent with the abbreviation, it seems in random order.   

11.  Also care must be taken for hour or h, min or minute as in line 129.

12.  Most of the methods start with “With a few adjustments”” first of all authors should not repeat the same term again and again. Secondly, modification should be mentioned.

13.  What is PDS (line 150.

14.  Line 155, sentence should not be starts with number.

15.  Line 164: From what concentrations, MP solution was diluted to 10 mg/mL.

16.  SDS-Page section is very weak, need to rewrite for better understanding. Samples were heated at 90C, with what or what concentration? No information.

17.  Section 2.4.9. When did authors make gels?? No method is given for the sample. Anyways, the author should group functional properties in one section and characterization in another.

18.  For the result and discussion section, no data is given during the discussion. What was the yield, how much it increased, should be given. It is easy to get data form the tables, but for bar charts the author must given the values for ease in reading understanding. Similar editing should be performed with other sections.

19.  The authors used statistical analysis, but I could not find citation anywhere. What was the significance (0.05 or 0.01). authors need to point out, whenever they make statement such as line 216234-235 and many more.

20.  Each section must be compared with the previous one, such emulsion properties with particle size or hydrophobicity.

21.  Section 3.3 and 3.4 should be compared with available literature.

22.  Line 252-256, to long and confusing sentence.

23.  Hydrophobicity must be compared with the turbidity, hydrophobic and hydrophobic interactions can leads to the aggregation, which should increase the turbidity, but authors have contrast results. Explained this.

24.  Line 306 has tryptophan and then line 307 has Trp, be consistent.

25.  Figure 4 should be enlarged. FTIR data is not visible at all.

26.  3.9 section should be near its citation. Please combine as suggested earlier.

27.  Provide the SDSpage image with wells and dye front. There is low intensity in the 500 W sample, but authors mentioned in contrast. This data should be supported by a densitometer.

28.  Line 421: name should be in italics.

29.  Conclusion is very poor and long. Explain main finding and suggest the future ideas.

30.  Somewhere authors used Figure and another time Fig or Fig. Check throughout the MS. Line 198, 323 or 346.

31.  References are not consistent. Line 458 or 474. Need to cross check for italic name and each uppercase letter in title.

Comments on the Quality of English Language

Too long sentences.

Usage of repeated words.

Reviewer 2 Report

Comments and Suggestions for Authors

The comments are in the attached file.

Comments on the Quality of English Language

The comments on the quality of English language are given in the review. 

Round 2

Reviewer 1 Report

Comments and Suggestions for Authors

All queries have been addressed satisfactorily, except for the following. After making these changes, the manuscript will be ready for acceptance

1. Remove (T. Molitor) from title, it is repeated.

Section 2.4.9. When did authors make gels?? No method is given for the sample. Anyways, the author should group functional properties in one section and characterization in another.

****I was talking about the SEM gels, in this MP gel sample were made. But I can not see the method of the same. Please explain the method for MP gel preparation (what was the concentrations of the sample, setting temp etc.)
